# Combination of St. John’s Wort Oil and Neem Oil in Pharmaceuticals: An Effective Treatment Option for Pressure Ulcers in Intensive Care Units

**DOI:** 10.3390/medicina59030467

**Published:** 2023-02-27

**Authors:** Samet Özdemir, Saber Yari Bostanabad, Ayhan Parmaksız, Halil Can Canatan

**Affiliations:** 1Department of Pharmaceutical Technology, Faculty of Pharmacy, Istanbul Health and Technology University, Istanbul 34015, Turkey; 2Department of Pharmacology, Faculty of Pharmacy, Istanbul Health and Technology University, Istanbul 34015, Turkey; 3Department of Biostatistics, Faculty of Medicine, Istanbul Health and Technology University, Istanbul 34015, Turkey; 4Intensive Care Unit, Department of Anesthesiology and Reanimation, Istanbul Special Güngören Hospital, Istanbul 34164, Turkey

**Keywords:** St. John’s wort oil, Neem oil, pressure ulcers, herbal ointment

## Abstract

*Background and Objectives*: Phytotherapeutically, various herbal remedies, such as St. John’s wort oil, have been introduced as wound care options. Recently, Neem oil has been considered a herbal option for the management of superficial wounds. Wound care is a complex process that involves several factors including the patient, caregiver, and medications. Herbal combinations could be an alternative to the chemical counterparts in the wound care area. This report includes an investigation of the possible supportive impacts of the St. John’s wort and Neem oil containing ointment (W Cura G Plus ^®^) in the management of pressure ulcers (PUs) in three intensive care unit (ICU) patients. *Materials and Methods*: The ointment was administered to individuals once daily for 42 consecutive days. The status of individuals was macroscopically monitored by measuring the PU area and histopathological assessment of the tissue sections taken on the first and last days of wound treatment. *Results*: The outcomes of the macroscopic and histopathological techniques exhibited that St. John’s wort and Neem oil containing ointment provided a remarkable supportive impact on the patients that suffered from PUs in the ICUs. *Conclusions*: The combination of St. John’s wort and Neem oil could be suggested as an efficient active phytoconstituent for the management of PUs. The herbal ointments may be suggested as an alternative for the patients that have PUs in the ICUs.

## 1. Introduction

Pressure ulcers (PUs) are a remarkable problem for patients and health professionals. This adverse event prolongs the time of hospitalization, and elevates mortality rates and medical expenses. Terminologically, different definitions have been used to express pressure-induced wounds, such as decubitus ulcer, pressure sore, and bedsore. The National Pressure Ulcer Advisory Panel (NPUAP) announced current guidelines, revising the preferred term as ‘pressure injury’ to consider all types of tissue damage induced by pressure [1]. A PU is described as local damage of the dermal region and inferior layers, usually over a bony prominence, or related to a medical or other device as a result of dense and/or extended pressure or friction [2]. The bony prominences of the sacral region and lower extremities are reported as the most frequently affected areas [2,3].

Clinically, the prevalence of PUs is high in patients that are hospitalized in intensive care units (ICUs). Several studies have expressed the high prevalence rates of PUs in the ICU setting [4,5]. Specific risk factors have been reported as advanced age, sex, impaired microcirculation, instability, etc. [2,6]. Proactive approaches are beneficial for the prevention of PUs. Several risk assessment scales have been developed and include various parameters for calculating the degree of risk [6]. The Braden Scale is recommended for ICU patients due to express the relationships between PU risk, nursing workload, and degree of illness [6]. Although prevention is the primary approach (using pressure-reducing support surfaces, repositioning, adequate nutrition, etc.) for PUs, management strategies include: wound-care fundamentals (cleansing debridement, infection control, and dressing selection), nonsurgical therapies (topical agents, negative pressure therapy, tissue-based products, and hyperbaric oxygen), and surgical management [3].

Herbal remedies may be an alternative option for the management of PUs. Traditionally, the flowering parts of St. John’s wort (*Hypericum perforatum* L.) have been used for the topical treatment of wounds, cuts, and burns [7,8]. The oily extracts are selected for the management of wounds due to their occlusive potential on the wound surface. Additionally, the oily extract contains lipophilic phytoactive compounds (hypericin and hyperforin) that potentiate the wound healing. The wound healing mechanism is based on the stimulation of collagen production, enhancement in fibroblast motility, and increased keratinocyte differentiation [9]. Topical drug delivery systems provide an efficient transfer of bioactives to the wound area. For instance, a St. John’s wort oil loaded ointment formulation exhibited a significant wound healing effect with a high safety profile [10]. In some cases, sophisticated delivery systems may be an alternative for the delivery of phytoconstituents, e.g., the standardized St. John’s wort extract containing niosomal gel formulation produced a significant reduction in wound size [11]. The combination of St. John’s wort, sage, and oregano oils in a single product was revealed as a striking example due to the sage and oregano oils potentiating the wound-healing effect [12].

Another action for PU therapy could be the Neem tree (*Azadirachta indica* L.) due to its reported wound healing activity [13,14,15]. Wound-healing activity was shown on mice by using excision and incision wound models [13]. In another study, a systemic Haridra (*Curcuma longa* Linn.) capsule application with subsequent topical Neem oil application was reported for the treatment of nonhealing wounds [14]. In that study, the synergistic effect was observed; however, single use of Neem oil was comparatively better than the single use of Haridra capsules. In an uncontrolled study, Neem oil and St. John’s wort oil loaded wound dressing were investigated on wound myiasis in domestic animals [15]. In that study, phytochemical content of Neem (azadirachtin a and b, nimbin, salannin, and gedunin) was reported to be antiparasitic given effects on larvae causing myiasis with low toxicity; thus, a synergistic effect was observed with the St. John’s wort oil on the wound healing process.

The goal of this clinical study was to investigate the potential effect of St. John’s wort and Neem oil containing a topical ointment preparation for the treatment of PUs in ICU patients.

## 2. Materials and Methods

### 2.1. Medication Material

In clinical case studies, a market product (W Cura G Plus^®^, Fitolab, Istanbul, Turkey) was administered for PU management due to its herbal content. The market product is a conventional ointment that has standardized St. John’s wort oil and Neem oil in its content. The formulation was prepared by using a nonionic surfactant mixture at low concentrations (<2.5%, w/w) to obtain an oil in water (o/w) type of ointment base. The herbal oils (>0.1%, w/w) were simultaneously emulsified while the process was in progress.

### 2.2. Medical Case Study

W CURA G Plus^®^ was administered to the three patients who had been admitted to the ICU. The patient consent declaration was taken from a first-degree relatives. 

#### 2.2.1. Patient I

Patient I was an 84-year-old male patient who had been bedridden for two years. Prediagnostic findings were as follows: heart failure, successful resuscitation after cardiac arrest, epilepsy, encephalopathy, respiratory failure, pneumonia, and sepsis. Patient was admitted to the ICU in October 2022.

The general health status of the patient was assessed as poor, unconscious (Glasgow Coma Scale: 8), and uncooperative. The patient was subjected to tracheotomy and was supported with a respiratory apparatus after being taken to the ICU.

According to the EPUAP chart, the patient had a stage-3 PU wound with dimensions of 9.4 × 7.5 cm^2^ on the sacrococcygeal area. Additionally, the Braden Risk Assessment Scale was applied for an evaluation of the risks for the patient’s PU. According to the risk assessment scale, the patient’s PU score was found to be 9.

#### 2.2.2. Patient II

Patient II was a 70-year-old male patient who had been bedridden for one year. Prediagnostic findings were as follows: epilepsy, cardiopulmonary successful resuscitation, diabetes mellitus, hypertension, pneumonia, and respiratory failure. The patient was admitted to the ICU in October 2022.

The general health status of the patient was assessed as poor, unconscious (Glasgow Coma Scale: 9), and uncooperative. The patient was subjected to tracheotomy and was supported with a respiratory apparatus.

According to the EPUAP chart, the patient had a stage-2 PU wound with dimensions of 7.4 × 6.1 cm on the sacrococcygeal area. Additionally, the Braden Risk Assessment Scale was applied for an evaluation of the risks for the patient’s PU. According to the risk assessment scale, the patient’s PU score was found to be 9.

#### 2.2.3. Patient III

Patient III was a 79-year-old man who had had chronic asthma for over five years. The prediagnostic findings were as follows: cardiopulmonary successful resuscitation, diabetes mellitus, pneumonia, and respiratory failure. The patient was admitted to the ICU in July 2022.

The general health status of the patient was assessed as poor, unconscious (Glasgow Coma Scale: 9), and uncooperative. The patient was subjected to tracheotomy and was supported with a respiratory apparatus.

According to the EPUAP chart, the patient had a stage-2 PU wound with dimensions of 4.6 × 6.6 cm on the sacrococcygeal area. Additionally, the Braden Risk Assessment Scale was applied for an evaluation of the risks for the patient’s PU. According to the risk assessment scale, the patient’s PU score was found to be 9.

### 2.3. Patient Consent

Before the pharmaceutical applications, the relatives of the patient were informed. A form declaring the consent of the patient was accepted and signed by a first-degree relative of the patient.

### 2.4. Treatment Approach and Monitoring

A wound hygiene protocol was followed for PU care [16]. The PU area was cleaned with normal saline solution before each application. The necrotic tissue or debris was removed by using a surgical knife. Then, overhanging wound edges were aligned to provide refashioning. After that, W CURA G Plus^®^ was administered to the patients once a day for 42 consecutive days (between 28 July and 7 September 2022, 10 October and 20 November 2022, and 14 October and 24 November). Additionally, the PU area was covered with a sterilized gauze pad after each W CURA G Plus^®^ application. Images of the PU area were obtained to follow the healing process for predetermined time intervals.

### 2.5. Histopathological Examination

Tissue samples from the PU area borders were removed on the initial day before the treatment (28 July 2022, 10 October 2022, and 14 October 2022) and on the last day (7 September 2022, 20 November 2022, and 24 November 2022) just before the final application. Tissue samples were obtained by the incisional biopsy technique. Histopathological sections were stained with hematoxylin and eosin-Y and images were examined under a light microscope (Olympus BX 51 TF, Olympus, Tokyo, Japan).

### 2.6. Statistical Analysis

For numerical variables, the mean and standard deviation are given as descriptive statistics. To investigate the treatment effect, the evaluations of the patients who were followed for 6 weeks were analyzed by repeated measures ANOVA. When the overall test result was found to be significant, a simple-contrast test was performed to determine at what time point statistical significance emerged according to the initial evaluations. We report the results in estimated mean difference and standard error of the estimated mean difference. A *p*-value < 0.05 was considered significant. Statistical analysis was performed by using JASP (JASP Team, University of Amsterdam, Version 0.16.1, Amsterdam, The Netherlands).

## 3. Results

### 3.1. Histopathological Assessment

The treatment regimen was provided to the patients for 42 consecutive days. The ointment was applied once a day. The histopathological samples were withdrawn from the patients at the beginning (day 1) and at the end of the treatment (day 42). The histological images are presented in Figure 1. 

### 3.2. Before the Medication

Acute inflammation was observed in the biopsy sample taken from patient I before the treatment. Histologically, neutrophils, stratified squamous epithelium, cells with small nuclei, medium cytoplasm, and usual chromatin material were present. Nucleolus was prominent, usual granular layer, usual keratin material, and basal vascularization was present.

Scattered inflammation and neutrophils associated with acute inflammation were detected in the biopsy sample obtained from patient II before the treatment. Histologically, stratified squamous epithelium; fibrinopurulent exudate; foreign body; small nuclei from squamous epithelium cytoplasm; and small, prominent nucleoli were detected.

Acute inflammation, neutrophils, and the usual stratified squamous epithelial tissue were observed in the biopsy sample collected from patient III before the treatment. Histologically, small nucleated cytoplasm, small nucleolus, usual granular layer, and keratin material were present.

### 3.3. After the Medication

Hyperplastic stratified squamous epithelium and hyperkeratotic stratified squamous epithelium were observed in the biopsy sample taken from patient I after treatment (42nd day). Histologically, nuclei became larger, nucleoli became more prominent, cytoplasm was high, granular layer thickened, and subepithelial diffuse capillary vascularization was detected. The histopathological outcomes of patient I were interpreted as tissue regeneration at the end of the treatment period.

As stated for the previous patient (patient I), hyperkeratotic stratified squamous epithelium and hyperplastic stratified squamous epithelium were observed in the biopsy sample taken from patient-II after the treatment. Histologically, nuclei became larger, nucleoli became more prominent, the cytoplasm was moderate, and subepithelial diffuse capillary vascularization was found. The epithelial appearance was assessed as complete tissue regeneration.

As in the previous two patients, from the biopsy sample taken from patient III, hyperkeratotic stratified squamous epithelium, hyperplastic stratified squamous epithelium were detected. Nuclei became larger, nucleoli became more prominent, cytoplasm was moderate, and subepithelial diffuse capillary vascularization and an epithelial appearance compatible with regeneration were present. The details of healing process were presented in Table 1.

Additionally, the wound-healing process was followed by using Wound Bed Score (WBS) [17]. This system assesses the healing trajectory including: healing edges, black eschar, wound depth/granulation, exudate amount, edema, periwound dermatitis, peri-wound callus/fibrosis, and pink wound bed [17,18]. Each criterion is scored between zero (worst condition) and two (best condition). The obtained scores are added; the sum of scores varies from 0 (undesirable score) to 16 (most desirable score). The sum of scores are divided into four quarters: up to 9, 10 or 11, 12 or 13, and 14 to 16. For each quartile increase, there is an increased chance of healing. The weekly assessment of patients is presented in Table 2.

### 3.4. Results of Statistical Analysis

A repeated measures ANOVA was performed to compare the effect of treatment on the stage of PUs, Braden score, and dimension of PU area for 6 weeks of follow-up.

The six-week treatment program had a statistically significant effect on the stage of the PU (F = 13.878, *p* < 0.001, η²_p_ = 0.874). A statistically significant difference was observed on the 25th day compared with simple-contrast comparisons with the baseline level (Table 3).

The six-week treatment program did not have a statistically significant effect on the Braden score, but the effect size of the treatment appeared to be very large (F = 2.161, *p* = 0.078, η²_p_ = 0.519).

The six-week treatment program had a statistically significant effect on the dimension of PU area (F = 18.861, *p* < 0.001, η²_p_ = 0.904). A statistically significant difference was observed on the 10th day compared with simple-contrast comparisons with the baseline level (Table 4).

The six-week treatment program had a statistically significant effect on the wound bed score (WBS) (F = 17.322, *p* < 0.001, η²_p_ = 0.896) (Table 5). A statistically significant difference was observed on the second week compared with simple-contrast comparisons with the baseline level (Table 6).

## 4. Discussion

With the expansion of public interest in herbal medicines, research has been conducted on various plants, and it is predicted that, soon, these medicinal plants will account for a major share of treatments [19]. Accordingly, there is growing evidence that St. John’s wort (*Hypericum perforatum*) and Neem (*Azadirachta indica*) have therapeutic potential and uses in herbalism and modern medicine [20,21,22].

St. John’s wort is a flowering plant that was reported to treat depression or physical disorders with related symptoms such as anxiety or insomnia, and was proposed as an anticancer agent, as well as to possess antimicrobial and anti-inflammatory activities [9,23]. Furthermore, it has been reported show some clinical and pharmacological activities such as stimulating collagen production, fibroblasts migration, and keratinocyte differentiation, and improving epithelialization, granulation, and wound healing [9,24,25].

St. John’s wort has shown potential for use as a wound healing treatment for various skin conditions, including wounds, pressure ulcers, bruises, and muscle pain, as a topical treatment (applied to the skin) [9]. Wound healing studies conducted by employing St. John’s wort demonstrated the ability of the plant extract and essential oil to converge wounds and reduce the time of epithelization, showed efficacy on cell migration and proliferation, and acted through antimicrobial and anti-inflammatory activities, leading to tissue repair [26].

The other plant used in the study was *Azadirachta indica*. It is an evergreen tree that is one of the important medicinal plants in the history of humankind [27]. Neem was traditionally employed for teeth cleaning, smeared skin disorders, and illnesses such as smallpox and infectious diseases in various regions of the world such as America, Africa, Persia, and India [28].

In 1992, the U.S. National Academy of Sciences defined ‘Neem—a tree for solving global problems” by outlining its magnificent properties. Moreover, the U.S. Environmental Protection Agency (US EPA) has approved the use of neem products in crops as it is safer for humans, insects, and animals [29].

Recent studies demonstrated A. indica has an effective position in disease management by regulating several biochemical signaling pathways and other biological processes [30]. Furthermore, several biological and pharmacological activities of Neem compounds have been reported, such as antibacterial activities, antioxidant, anti-inflammatory, antiarthritic, antipyretic, antiviral, spermicidal, hypoglycemic, anthelminthic, antigastric ulcer, and antitumor activities (several studies have demonstrated the anticancer activity of nimbolide in various cancer cells) [15,29,30,31].

We carried out this study in the continuation of a previous study in which we investigated the effect of St. John’s wort on the care and treatment of pressure sores (wound healing). At the end of the previous study we demonstrated that St. John’s wort oil has treatment potential (topical administration) for pressure ulcers. Considering the previous work and other recent studies, especially studies about the Neem plant, we determined that each of these therapeutic herbs has a valuable role in wound healing. Hence, we designed this study: rather than using each of these therapeutic plants separately, we employed a combination of these herbs for the treatment of wounds.

The present study explains our clinical examination, demonstrating the usage of a combination of St. John’s wort oil and Neem oil in three patients with PUs in ICUs. At the end of the investigation, we found that all pressure ulcers eventually healed after a mean healing time of 42 days. This report includes three cases of people who had pressure ulcers in the ICU with different severities. The treatment was administered to the PU patients once a day nonstop for 42 consecutive days. Wound healing and ulcer size were recorded and microscopically assessed each day. Consequently, complete wound healing could finally be observed in all PU patients with treated ulcers. Additionally, we were able to obtain wound healing results, which we saw as a result of macroscopic trials in the histopathologic examination. In the study, we used a combination of Neem and St. John’s wort oil as a topical administration and ointment formulation.

In line with our results, the treatment of peristomal wounds with a combination of Neem and Red Hypericum oil as a topical application was reported. For this study, two patients were allocated and treated twice weekly and had complete healing of the wounds in one of the cases after 14 days and in the other case after 28 days [32].

*Hypericum perforatum* and *Azadirachta indica* (Neem) oils were applied to calcinosis-related ulcers in systemic sclerosis. Our results are found in agreement with those of that study. The aforementioned work demonstrated complete wound healing in 45% of cases within a period of 40.1 ± 16.3 days, while 55% of ulcers improved in terms of size, fibrin, erythema, and calcium deposits [33].

A recent study that correlates with our investigation was performed on 16 chronic leg ulcer patients using treatment with a plant-derived wound dressing based on *Hypericum perforatum* and *Azadirachta indica* (Neem) oils. At the end of the study, all ulcers demonstrated complete healing with a median healing time of 82 days [34].

In this study, the healing grade was macroscopically monitored by measuring the wound size, stages, and Braden scores at certain intervals as well as histopathological evaluation at the initial and the final dates of treatment. Histopathological studies demonstrated their potential for tissue regeneration and progression of healing in the pressure sore area with the development of dermis/epidermis and collagen. These supported our macroscopic results in pressure wound healing. To the best of our knowledge, this study is the first to demonstrate the effect of a combination of Neem and St. John’s wort on wound healing with the use of a histopathology method. However, there is some evidence and histopathological outcomes that these herbs can separately heal wounds. Some studies reported in vivo wound healing with *Hypericum perforatum* ointment, as evaluated with histopathology, in wound treatment models in Wistar rats.

The histopathological examination of the affected areas that were treated for 21 days with Hyperici herbal ointment revealed the presence of mature granulation tissue in almost all the depths of the dermis for the incision and excision wound models. The clinical and histopathological results, along with the wound contraction rate and period of epithelialization, demonstrated the wound-healing effect of the novel St. John’s wort ointment in linear incisions, circular excisions, and thermal burns [10].

Another study reported wound healing on Wistar albino rats using histopathologic examination and an *Azadirachta indica* (Neem) oil topical formulation. In this study, histology sections from the 10th and 20th postwounding days showed re-epithelialization and formation of dermis and epidermis along with collagen synthesis in the treated animals. These results are comparable to those obtained with standard diclofenac gel and neomycin cream, which are currently employed for inflammation control and wound healing [35].

## 5. Conclusions

A combination of Neem oil and St. John’s wort oil in a single pharmaceutical preparation could be a natural remedy for PUs. After 42 days of follow up, the full thickness healing of the skin was achieved by using a natural remedy; thus, it can be concluded that these remedies could be an effective pharmaceutical alternative for wound healing.

## Figures and Tables

**Figure 1 medicina-59-00467-f001:**
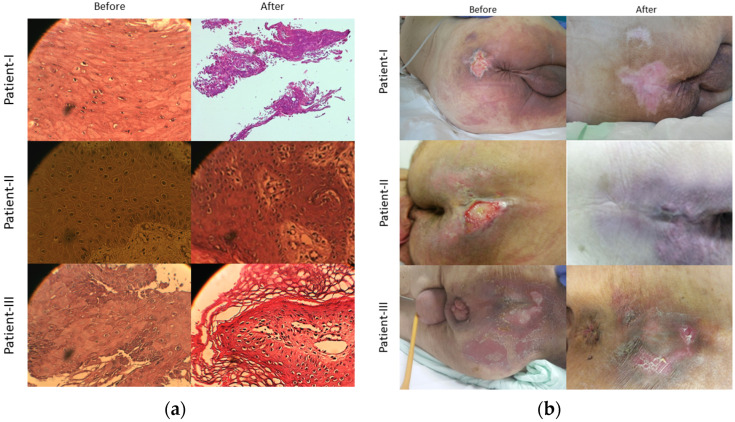
Images were taken on the sacrococcygeal area and represent the histological condition of the PU (**a**). Visual observation of wound area (**b**).

**Table 1 medicina-59-00467-t001:** Evaluation chart of the pressure ulcer (PU) of patients during the treatment regimen.

	Patient I			Patient II			Patient III		
Day	Stage of PU	Braden Score	Dimensions of PU Area (cm)	Stage of PU	Braden Score	Dimensions of PU Area (cm)	Stage of PU	Braden Score	Dimensions of PU Area (cm)
1st	3	9	9.41 × 7.53	2	9	7.42 × 6.08	2	10	4.61 × 6.63
5th	3	9	7.80 × 6.90	2	9	7.38 × 5.06	2	10	4.42 × 6.21
10th	3	10	7.58 × 6.01	2	10	4.80 × 4.36	2	10	4.13 × 5.34
15th	2	10	6.40 × 4.78	2	10	4.28 × 3.43	1	11	3.94 × 4.86
20th	2	11	4.73 × 4.24	2	11	2.90 × 2.68	1	16	2.43 × 4.50
25th	2	11	4.21 × 3.95	1	11	2.88 × 1.77	Full thickness	18	1.92 × 3.01
30th	2	11	3.72 × 3.03	1	11	2.12 × 1.13	Full thickness	21	1.40 × 2.69
35th	2	11	3.48 × 2.50	1	11	1.88 × 1.09	Full thickness	22	1.15 × 2.28
40th	1	11	2.56 × 1.48	Full thickness	11	1.85 × 0.92	Full thickness	22	1.03 × 1.50
42nd	Full thickness	11	1.44 × 0.93	Full thickness	11	1.80 × 0.64	Full thickness	22	1.00 × 1.46

**Table 2 medicina-59-00467-t002:** WBS of the patients.

	WBS (Weekly Follow-Up)						
	Initial	Week 1	Week 2	Week 3	Week4	Week 5	Week 6
Patient I	6	9	12	13	14	16	16
Patient II	6	9	12	13	14	16	16
Patient III	10	12	14	15	15	16	16

**Table 3 medicina-59-00467-t003:** Time-based comparisons of stage of PU, Braden score, and dimension of PU area.

	Stage of PU	Braden Score	Dimension of PU Area
Day 1	2.33 ± 0.58	9.33 ± 0.58	48.85 ± 20.4
Day 5	2.33 ± 0.58	9.33 ± 0.58	39.54 ± 13.32
Day 10	2.33 ± 0.58	10 ± 0	29.51 ± 13.91
Day 15	1.67 ± 0.58	10.33 ± 0.58	21.47 ± 8.21
Day 20	1.67 ± 0.58	12.67 ± 2.89	12.92 ± 6.38
Day 25	1 ± 1	13.33 ± 4.04	9.17 ± 6.47
Day 30	1 ± 1	14.33 ± 5.77	5.81 ± 4.78
Day 35	1 ± 1	14.67 ± 6.35	4.46 ± 3.69
Day 40	0.33 ± 0.58	14.67 ± 6.35	2.35 ± 1.25
Day 42	0 ± 0	14.67 ± 6.35	1.32 ± 0.16
F	13.878	2.161	18.861
*p*	<0.001	0.078	<0.001
η²_p_	0.874	0.519	0.904

η²_p_: Partial Eta Square (effect size).

**Table 4 medicina-59-00467-t004:** Simple contrast results for the stage of PU, Braden score, and dimension of PU area.

	Stage of PU		Dimension of PU Area	
Comparison	Est ± SE	*p*	Est ± SE	*p*
Day5–Day1	0 ± 0.32	1.000	−9.31 ± 5.45	0.105
Day10–Day1	0 ± 0.32	1.000	−19.33 ± 5.45	0.002
Day15–Day1	−0.67 ± 0.32	0.051	−27.37 ± 5.45	<0.001
Day20–Day1	−0.67 ± 0.32	0.051	−35.92 ± 5.45	<0.001
Day25–Day1	−1.33 ± 0.32	<0.001	−39.68 ± 5.45	<0.001
Day30–Day1	−1.33 ± 0.32	<0.001	−43.03 ± 5.45	<0.001
Day35–Day1	−1.33 ± 0.32	<0.001	−44.39 ± 5.45	<0.001
Day40–Day1	−2 ± 0.32	<0.001	−46.5 ± 5.45	<0.001
Day42–Day1	−2.33 ± 0.32	<0.001	−47.53 ± 5.45	<0.001

Est ± SE: Estimation of mean difference and standard error.

**Table 5 medicina-59-00467-t005:** Time-based comparisons of the WBS.

	Wound Bed Score (WBS)
Initial	7.33 ± 2.31
Week 1	9 ± 3
Week2	11 ± 3.61
Week 3	13 ± 2
Week 4	13.67 ± 1.53
Week 5	15.33 ± 1.16
Week 6	16 ± 0
F	17.322
*p*	<0.001
η²_p_	0.896

**Table 6 medicina-59-00467-t006:** Simple contrast results for the WBS.

	WBS	
Comparison	Est ± SE	*p*
Week 1–Initial	1.67 ± 1.1	0.154
Week 2–Initial	3.67 ± 1.1	0.006
Week 3–Initial	5.67 ± 1.1	<0.001
Week 4–Initial	6.33 ± 1.1	<0.001
Week 5–Initial	8 ± 1.1	<0.001
Week 6–Initial	8.67 ± 1.1	<0.001

Est ± SE: Estimation of mean difference and standard error.

## Data Availability

The datasets used and/or analyzed in the current study are available from the corresponding author upon reasonable request.

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
