# Peer review of "Combination of St. John’s Wort Oil and Neem Oil in Pharmaceuticals: An Effective Treatment Option for Pressure Ulcers in Intensive Care Units"

_medicina, 2023, doi:10.3390/medicina59030467_

Round 1
Reviewer 1 Report
In accordance with the Wound Hygiene procedure (https://doi.org/10.12968/jowc.2022.31.Sup4a.S1), the Authors indicated how pressure wounds were cleaned (step 1) and dressed (step 4). I recommend clarifying how were carried out the intermediate steps of debridement (step 2: removal of necrotic tissue, slough, debris, and biofilm at each dressing change) and wound edge refashioning (step 3: remove necrotic, crusted and/or protruding wound edges that may harbor biofilms. Ensure skin edges are aligned with the wound bed to facilitate advancement and contraction of the epithelium). In accordance with the TIMERS procedure (DOI: 10.12968/jowc.2019.28.Sup3a.S1), I recommend clarifying the healing trajectory, with preferably weekly assessment, of the Wound Bed Score (necrosis, fibrin or granulation); any signs of infection and how it was treated; the amount of exudate; the area of the wound (square centimeters or inches); any other therapies applied to facilitate pressure injury repair.
Author Response
Dear reviewer,
We would like to thank you for this valuable contribution; A standard wound care was applied for the PU in the ICU. But we would like to give brief information about wound care.
The procedure (steps 2 and 3) was explained in detail in section “2.4. Treatment approach and monitoring”. Then, the recommended procedure was also cited.
A previously published (Falanga et. al. introduced) scoring system was applied. WBS was calculated by using the information from the patient’s file and the images of the wound area. The sum of scores was shared and statistics were applied. The related citations were also given.
Best regards.

Reviewer 2 Report
Samet Özdemir et al., submitted a paper. Though the paper was well written, it needs to rectify the following things
1. Abstarct should start with a brief introduction.
2. Need ethical clearance
3. Need some more information on the formulation of ointment
4. Only three patients' information was provided......feeling that we may need more sample size.
5. Need more statistical interpretation
Author Response
Dear Reviewer,
We would like to thank your contribution and comments. Hereby, we would like to present our responses.
1. Abstarct should start with a brief introduction.
We would like to thank your kind contribution. A brief introduction was added to the “Background and Objectives” part of the abstract. The revised parts could be easily detected due to the tracking mode of MS Word.
2. Need ethical clearance
The marketed product has been already approved and is available in the Turkish market with type IIb medical device status.
Additionally, the medical center (Istanbul Special Güngören Hospital) follows the Johns Hopkins Medicine’s Case Report Publication Guidance: IRB Review and HIPAA Compliance (Policy No. 102.3) – (https://www.hopkinsmedicine.org/institutional_review_board/guidelines_policies/organization_policies/102_3.html)
Ethical review and approval were waived for this study, as the cases reported describe the standard of care in the involved hospitals.
But we provide informed consent forms from the patient’s relatives. You can find these forms as supplementary files.
3. Need some more information on the formulation of ointment
The formulation is a confidential part of the marketed products. When we reached the company, they only shared generalized information that we have inserted in the “2.1. Medication material” section.
The additional part could be easily detected due to the tracking mode of MS Word.
4. Only three patients' information was provided......feeling that we may need more sample size.
We would like to thank the reviewer. The current study was planned as a case report study.
Generally, case reports have a smaller group of patients (The number of participants can vary from one to ten) than original research articles.
We select the maximal (3 subject) amount of patient that the policy allows us (https://www.hopkinsmedicine.org/institutional_review_board/guidelines_policies/organization_policies/102_3.html).
Hereby, we would like to show some examples:
- https://doi.org/10.3390/medicina59020225
- https://doi.org/10.3390/medicina57030296
- https://doi.org/10.1016/j.jep.2016.12.030
5. Need more statistical interpretation
Professional support was taken and a specialist was added as an author for his valuable contribution. Statistical analysis and interpretation were performed. Related information was also embedded in the text.
Sincerely yours.

Reviewer 3 Report
The authors of the presented study demonstrate impact of combination treatment of St John´s wort and Neem oil containing ointment on pressure ulcers in three bedridden intensive care unit patients. To their best knowledge, its the first time such combination treatment was used, although reports of separate/single treatment outcomes already exist. The results of the presented case report clearly demonstrate high potential of combination of St John´s Wort and Neem oil treatment in patients with pressure ulcers. In my humble opinion, further single treatment and combination treatment results are neccessary to clearly validate the pharmaceutical potential of the presented treatment. To sum it up, the data presented in this manuscript meet conditions of Case report and represent promising strategy for future treatment of pressure ulcers.
Please consider following points to further improve quality of the manuscript (optional):
1. Line 247 - english correction
2. Figure 1 - Can the pictures be stretched to better fit the page?
3. Acknowledgements section - the section seems not filled
4. Appendix A - the section seems not filled
Author Response
Dear reviewer,
We would like to thank your contributions and comments. Hereby, we would like to present our responses.
1. Line 247 - english correction
The correction was achieved. The revised part could be easily detected due to the tracking mode of MS Word.
2. Figure 1 - Can the pictures be stretched to better fit the page?
The template of this manuscript did not allow stretching. This condition has been accepted as part of the manuscript’s format.
3. Acknowledgements section - the section seems not filled
We would like to thank the critical advice for the reviewer. An acknowledgement was added to the related part.
The revised part could be easily detected due to the tracking mode of MS Word.
4. Appendix A - the section seems not filled
The supplementary data were given in the manuscript's “supplementary material” part. Thus, this section was removed.
Best regards,

Round 2
Reviewer 1 Report
Comments from the previous revision have been incorporated into the new version of the manuscript. Congratulations on your professional commitment.
Reviewer 2 Report
The authors made significant changes as per the comments made
Hence, I recommend for the publication in Medicina